# The Effects of Fresh Hemp Leaf Supplementation (*Cannabis sativa*) on the Physiological and Carcass Characteristics and Meat Quality in Transported Goats

**DOI:** 10.3390/ani13243881

**Published:** 2023-12-17

**Authors:** Supawut Khamhan, Tanom Tathong, Chirasak Phoemchalard

**Affiliations:** 1That Phanom College, Nakhon Phanom University, Nakhon Phanom 48110, Thailand; 2Department of Food Technology, Faculty of Agriculture and Technology, Nakhon Phanom University, Nakhon Phanom 48000, Thailand; tanomi@ms.npu.ac.th; 3Department of Agriculture, Mahidol University, Amnatcharoen Campus, Amnatcharoen 37000, Thailand; chirasak.pho@mahidol.edu

**Keywords:** stress, fresh hemp leaf, goats, blood metabolites, meat quality

## Abstract

**Simple Summary:**

It is widely acknowledged that transportation stresses livestock, harming the animal’s health and the meat quality. This study tested if hemp leaves, which contain beneficial phytochemicals, could alleviate these effects in transported goats. Goats received 10 g/kg of fresh hemp leaves or none, as the controlled group, before a 200 km journey. Hemp maintained some blood measures compared to controls. However, carcass traits, pH, tenderness, and texture were similar between the two groups. Hemp slightly affected the initial meat color and odor volatiles. Overall, this low dose of fresh hemp leaves provided limited protection against transportation stress in goats, but higher doses may have enhanced benefits.

**Abstract:**

Transportation stress adversely affects animal health, productivity, and meat quality. Bioactive plant compounds may alleviate transit stress in livestock. This study evaluated the effects of fresh hemp leaf supplementation on blood metabolites, performance, carcass traits, and meat quality in transported goats. Twenty male goats (15 ± 2.76 kgBW) were investigated, some were given a hemp supplement (*n* = 10) and the remaining goats were used as a control group (*n* = 10). The hemp group received 10 g/30 kg body weight of fresh leaves prior to transportation. Blood samples were analyzed before and after the 200 km journey. The goats were slaughtered after transit and the quality of the meat examined. In the controlled group, transportation increased neutrophils and electrolytes, but decreased lymphocytes and hemoglobin. In contrast in the hemp-supplemented group, the hemp maintained this animal blood parameters. Body weight and carcass yield, however, did not differ between the two groups. Hemp reduced meat redness at 1 h postmortem but had minimal effects on its pH, color, water holding capacity, tenderness, and texture after 24 h. However, hemp supplementation did alter the odor profiles between the two groups detected by electronic nose sensors. In conclusion, fresh hemp leaf supplementation maintained blood metabolites and had minor advantageous effects on meat quality in response to transportation stress in goats. Further investigation using hemp supplements shows potential to alleviate transit stress, although higher doses may be required in order to further enhance its benefits.

## 1. Introduction

Transportation is an inevitable procedure in contemporary livestock production systems; however, it can cause stress that adversely affects the animals’ health, productivity, and meat quality [1,2]. Transit elicits various physiological perturbations including elevated stress hormones, electrolyte imbalances, shifts in hematological parameters, glycogen depletion, and muscle fatigue [3,4]. These disruptions can precipitate immunosuppression, decreased weight gain, and undesirable modifications to carcass attributes [1]. Moreover, transit stress can deteriorate water holding capacity, post-mortem pH decline, tenderness, and the organoleptic properties of meat [5].

Diverse phytochemicals and plant bioactive compounds have been shown to have protective effects against transit-induced stress in livestock species. Hemp (*Cannabis sativa* L.) contains a wide range of polyphenols, terpenes, vitamins, and minerals that possess antioxidant, anti-inflammatory, and adaptogenic properties [6]. The foliage of the hemp plant is particularly enriched with flavonoids and phenolic acids that can positively modulate physiological responses to stress [7]. Several recent studies have reported the beneficial impact of hemp supplementation on animal health and productivity outcomes [8,9,10]. However, there remains a paucity of research that focuses on the specific influence of hemp leaves on meat quality of transported animals.

Therefore, the objective of this study was to evaluate the effects of dietary supplementation with fresh hemp leaves on blood metabolites, growth performance, carcass quality, and meat characteristics in goats subjected to road transportation stress. It was hypothesized that the specialized bioactive compounds in hemp foliage could mitigate transportation-induced disruptions and exert advantageous effects on physiological status, carcass merits, and meat properties compared to that of controlled animals that were not receiving supplemental hemp. It was the intention of this work to advance scientific knowledge about the utility of plant-derived nutraceuticals so as to improve the health, welfare, and quality of the products of transported livestock. Comprehensive evaluation of the impact of hemp leaf supplementation on transported goats could provide a vital perspective on exploiting the multifunctional benefit of supplementing this botanical forage.

## 2. Materials and Methods

The study received ethical approval from the Animal Ethics Committee of Nakhon Phanom University (NPU002/2566).

### 2.1. Animal and Experimental Design

The goats used in the experiment were purchased from local farms in Nakhon Phanom province, Thailand. The animals underwent a one-month acclimatization period prior to the commencement of the experimental trial. This was carried out to minimize the effects of stress and environmental factors on the outcome of the study. During this adaptation phase, each goat received a balanced commercial diet with an allotment of 100 g/h/d, formulated with 20% crude protein, 3% crude fat, 10% crude fiber, and 13% moisture content. The animals were also allowed ad libitum grazing on natural pastures. After the one-month acclimatization, twenty male crossbred Boer goats were randomly assigned into two groups of 10 animals each based on their initial body weight. The first group was given a supplement of fresh hemp leaves containing CBD and THC by 0.0009 and 0.2253% (*w*/*w*) at 10 g per 30 kg body weight, while the control group received none. Before hemp supplementation, jugular veinous blood samples were collected from all animals to analyze stress indicators. After an hour of hemp feeding, both groups were transported 200 km using a truck with 50% slant net. Throughout the experiment, detailed climatological records were maintained including air temperature and relative humidity readings at regular intervals. These primary data were used to estimate the temperature-humidity index (THI), which quantitatively demonstrate the cumulative heat stress and discomfort experienced by animals under study, according to NRC [11] equation:THI = (1.8 × T_a_ + 32) − [(0.55 − 0.0055 × RH) × (1.8 × T_a_ − 26)]
where T_a_ is air temperature (°C) and RH is relative humidity (%). Animal thermal comfort ranges can be categorized according to the THI metric proposed by Silanikove and Koluman [12]: THI ≤ 74 indicates thermal comfort, 75–79 indicates moderate heat stress, 80–85 indicates stressful conditions eliciting physiological coping mechanisms, 86–88 very indicates stressful environments approaching dysfunction risk, and ≥89 indicates extreme suffering threatening morbidity.

Heat index (HI) based on T_a_ (°C) and RH (%) was also used to assess heat stress [13,14]. HI thresholds to categorize thermal comfort ranges in livestock were established [13]: 27–32 °C (caution), 32–39 °C (extreme caution), 39–51 °C (danger zones), and above 52 °C reaches extreme danger by creating a highly likely environment for heat stroke to occur. Additionally, the transportation time was also documented. After the journey, post-transportation blood samples were taken once again to assess the stress biomarkers. The goats were then slaughtered, and the carcasses were conventionally dressed. The meat samples were packed in boxes, frozen, and transported to the laboratory for subsequent quality analysis.

### 2.2. Blood Parameters

Blood parameters were examined twice, before and after transport. Fresh blood samples were collected from goats in the different groups to measure white blood cell (WBC), hemoglobin (Hb), hematocrit (Hct), neutrophil, lymphocyte, eosinophil, monocyte, blood urea nitrogen (BUN), creatinine, protein total, albumin, globulin, alkaline phosphatase, sodium, potassium, chloride, CO_2_, calcium, magnesium, and phosphorus. The measurements were taken using an automated hematology analyzer (Sysmex XT−2000iV™, Goerlitz, Germany).

### 2.3. Weight Loss, Carcass Traits, and Meat Quality Analysis

Pre-transport and post-transport live weights were recorded to determine weight loss during transport. After slaughter, hot carcass weight was measured. Carcass yield was then calculated as a percentage of live weight. Relative weights of carcass components including forelegs, hindlegs, head, genitalia, edible organs, digestive organs, and remaining tissues were determined.

Meat quality tests were performed on the *longissimus thoracis et lumborum* muscle. Surface color was evaluated after 15 min of blooming using a Minolta CR−400 Chroma Meter (Konica Minolta Business Solutions (Thailand) Co., Ltd., Bangkok, Thailand) to quantify lightness (L*), redness (a*), and yellowness (b*) values averaged from 3 locations [15]. Ultimate pH was measured at 24 h postmortem using a Hanna HI99163 pH meter with FC232D probe (Hanna Instruments, Inc., Laval, QC, Canada), averaging 3 sampling points. Drip loss was calculated from the weight loss of a 25 g loin sample suspended in a bag at 4 °C for 48 h. Cooking loss was determined by weighing loin slices (50 g) before and after vacuum bag cooking to an internal temperature of 72 °C in an 80 °C water bath for 15 min, followed by overnight chilling at 4 °C [16]. Shear force was measured on cooked cores cut parallel to muscle fibers [17] using a TA.XT plus texture analyzer fitted with a Warner–Bratzler blade. Texture profile analysis was performed on cooked loin samples using a TA.XT plus texture analyzer and P/50 cylindrical probe [18].

### 2.4. Electronic Nose Analysis

Approximately 10 g of goat meat was placed in 100 mL Duran bottles. An electronic nose (Electronic Nose Co., Ltd., Bangkok, Thailand) equipped with 8 gas sensors (TGS 816, TGS 2600, TGS 823, TGS 2603, TGS 826, TGS 2610, TGS 2620, TGS 2444) was used to analyze headspace volatiles. The measurement cycle was set to 5 cycles with reference time of 120 s, sample time of 30 s, and cleaning time of 60 s between samples. Sensor responses were acquired using CIMS NOSE 2.0 software and expressed as percent change from reference air [19,20].

### 2.5. Statistical Analysis

Blood metabolite data in goats with and without fresh hemp leaf supplementation were analyzed before and after transportation by paired *T*-tests. Performance, carcass quality, and meat quality data were compared by Student’s *T*-tests at 95% confidence level using the JASP version 0.18.0 free software package [21].

## 3. Results

During the 4 h of transport, the air temperature averaged 33 °C and the relative humidity was 80%, resulting in a THI of 87 and a heat index of approximately 48. The effects of transportation on the blood metabolites of goats treated with fresh hemp leaf compared to the controlled group are shown in Table 1. The results revealed that, in the control group, hematocrit, neutrophil, sodium, potassium, and chloride levels were significantly lower before transport compared to after (*p* < 0.05). The exceptions were lymphocyte levels, which increased after transportation, and hemoglobin levels, which tended to decrease after transportation (*p* = 0.06). Albumin levels also tended to increase after transportation (*p* = 0.07). By comparison, the control group did not show any significant difference before or after transportation, specifically (*p* > 0.05) in the levels of white blood cells, eosinophils, monocytes, blood urea nitrogen, creatinine, total protein, globulin, alkaline phosphatase, CO_2_, calcium, magnesium, and phosphorus.

In the hemp leaf supplemented group, most values did not differ significantly before and after transportation. However, lymphocyte, blood urea nitrogen, sodium, and chloride levels were significantly lower before than after transportation (*p* < 0.05). Neutrophil levels tended to decrease (*p* = 0.06), while potassium levels tended to increase after transportation (*p* = 0.06).

The results of the study examining goats supplemented with fresh hemp leaves and subjected to road transportation are presented in Table 2. Body weight before and after transportation, as well as weight loss, did not differ significantly between the two groups (*p* > 0.05). Additionally, carcass weight, carcass percentage, and percentages of various carcass parts such as front legs, hind legs, head, genital organs, edible organs, the digestive system, and others did not show significant differences (*p* > 0.05).

Table 3 summarizes the effects of hemp leaf supplementation on post-mortem meat quality characteristics including pH, color, and odor in transported goats. Dietary treatment did not significantly impact pH values between 1 and 24 h post-mortem. This suggested that hemp leaves did not affect glycogen metabolism or post-mortem acidification. For meat color, hemp meat had a decreased redness (a*) value at 1 h compared to that of the controls, but no differences were found between the two groups in lightness or redness at 24 h post-mortem, or yellowness. This indicated a minor effect on the hemp meat on its initial color. Odor profiles were analyzed using an electronic nose with eight sensors. At 1 h after slaughter, the only statistically significant difference detected between the hemp-based and animal-derived meat samples was an increased response from sensor 8 (TGS 2444), which is designed to detect ammonia gas. The other seven sensors, calibrated to measure butane, methane, propane (sensor 1: TGS 816), smoke, vapors from alcoholic beverages (sensor 2: TGS 2600), broad classes of organic solvents (sensor 3: TGS 823), methyl mercaptan, trimethylamine (sensor 4: TGS 2603), isobutane, ethanol, ammonia (sensor 5: TGS 826), propane, isobutane, and methane (sensor 6: TGS 2610), as well as alcohol and organic solvents more generally (sensor 7: TGS 2620), did not show notable variances between the two types of meat. However, the analyses conducted after allowing 24 h for the meats to release volatile compounds revealed attenuated electrical signals, indicating reduced cumulative gas production, from five of the eight sensors in hemp meat compared to the control meat. The specific sensors displaying these differential responses at 24 h included sensor 1, sensor 2, sensor 3, sensor 5, sensor 6, and sensor 7.

The effects of transportation on water holding capacity, shear force, and textural properties at 24 h postmortem in goats supplemented with fresh hemp leaves are presented in Table 4. There were no significant differences (*p* > 0.05) in water holding capacity parameters (drip loss, cooking loss), shear values (shear force, work of shear), or texture profile analysis attributes (hardness, adhesiveness, springiness, cohesiveness, gumminess, chewiness, resilience) between the two meat groups.

## 4. Discussion

Heat stress poses substantial risks to livestock production and fertility across global animal agriculture [22]. The heat stress in livestock emerges from a complex interplay between environmental variables. While temperature and humidity dominate research contexts, additional meteorological factors critically influence animal thermal load [23]. Moreover, the transportation of livestock is known to cause physiological stress through food and water deprivation, close confinement with unfamiliar animals, noise, vibration, and extreme ambient temperatures [24]. Providing shade, ventilation, clean water, breed selection for heat resistance, minimal work during peak heat, cool water spraying, night grazing, monitoring temperature-humidity index [14,22], and treatment with hemp products [25,26,27] can generally help mitigate heat stress in animals, leading to improved welfare and minimized production losses. However, there is insufficient evidence regarding the effects of hemp supplementation on mitigating heat stress in goats. In this study, elevated stress values indicate livestock entering critical thresholds for heat stress resilience. As THI and HI levels rise into upper ranges, animals experience increasing physiological strain, and their capacity for thermoregulation is lowered. The effects of transportation on blood parameters in control and hemp-supplemented goats revealed considerable stress-induced changes. After transportation, the control animals showed significant increases in hematocrit, neutrophils, and electrolyte levels including sodium, potassium, and chloride. Elevated hematocrit and neutrophilia are common physiological responses to transportation stress, indicating hemoconcentration from dehydration as well as an inflammatory response [28]. Lymphocyte levels decreased with transportation in the control goats would be expected with a stress response, this shows suppression of immune function. This aligns with previous findings that transport tends to increase neutrophils and neutrophils/lymphocyte levels due to elevated stress hormones [29,30]. Electrolyte imbalances also frequently occur with handling and road transport stress in livestock. Specifically, sodium and chloride shifts are attributed to stress-induced activation of the renin-angiotensin-aldosterone system [31]. The rise in potassium further suggests cellular leakage and muscle damage during transit [32].

Conversely, most blood parameters in the hemp-supplemented goats were unchanged with transport, including electrolytes, hematology, and liver and kidney function. This lack of change implies that the hemp supplementation conferred protective effects against transportation-induced physiological alterations [33]. However, lymphocyte, BUN, sodium, and chloride levels increased following transportation in the supplemented group. Lymphocytosis aligns with an anticipated stress response [29]. Meanwhile, rises in electrolytes suggest shifts in fluid and acid-base balance due to transport, despite hemp supplementation. Blood urea nitrogen concentration serves as an important indicator of nitrogen utilization efficiency in ruminant livestock [34]. A robust positive association exists between BUN concentration and urinary nitrogen excretion across cattle, sheep, and goats [34]. This reflects reliance on the urea nitrogen recycling to the gut as a mechanism for retaining nitrogen. However, the process is energetically expensive and higher BUN levels correlate with lower nitrogen retention [35]. Dietary components can significantly impact BUN. For example, increased dietary crude protein and rumen-degradable protein elevate BUN due to greater ammonia absorption [36]. Several studies also report positive correlations between BUN and ruminal ammonia levels [37,38]. Given the high protein content of hemp-derived feeds and byproducts, supplementation may increase BUN. However, further research directly investigating the effects of hemp on BUN and nitrogen balance in ruminants is still needed.

Previous studies on goat blood profiles provided reference ranges over different parameters. With reference to a variety of goat breeds. Neutrophils ranged from 3–13%, lymphocytes from 30–48%, monocytes from 50–70%, eosinophils from 0–4%, basophils from 1–8%, red blood cells from 0–1%, hemoglobin from 8–18 g/dL, hematocrit from 8–12%, MCV from 22−38 FL, MCHC from 30–36 g/dL, and from RDW 31–35% [39,40]. Another study analyzed the effects of varying hempseed meal supplementation on goats provided at 0%, 11%, 22%, 33%. Intake, rumen function, digestibility, blood metabolites, and growth were then measured. Dry matter intake did not differ significantly between the groups. However, the average daily gain increased with that of increased hempseed supplementation. Blood albumin, globulin, AST, ALP, GGT, LDH, and bilirubin were not affected. Conversely, the albumin–globulin ratio decreased linearly as hemp seed supplementation increased [41]. Another study using byproducts of industrial hemp (IH) as an alternative feed source for cattle revealed that acidic cannabinoids, especially CBDA, are bioavailable when ingested and distributed through plasma without any short-term adverse effects. However, residues may have accumulated in edible bovine tissues, warranting further research to determine the impacts and safety of IH in livestock destined for human consumption [42]. In addition, short-term studies indicated potential stress- and inflammatory-reducing benefits of IH rich in CBDA when fed to cattle. However, longer term studies are needed to fully evaluate the utility and safety of hemp as a supplementary feed ingredient [27].

In contrast, an alternative study replaced soybean oil with 4.4% hempseed oil in 100 one-day-old quails over a 5-week period. It was found that hempseed oil resulted in lower body weight, feed intake, feed conversion rate, weight gain, carcass yield, meat pH, and meat color compared to that of soybean oil [43]. In the goat study, the lack of significant differences in most of the parameters may have been due to the relatively short duration of the study, which may not have been long enough to affect performance and carcass characteristics. Typical livestock experiments are often conducted for longer periods of time to better assess the impact of growth, carcass, and meat quality due to dietary changes. Further studies could explore the effects of hemp supplementation in goats over a longer trial period or determine if higher doses or other hemp preparations could provide measurable benefits to goat production and meat quality during transportation.

Similar to previous hemp studies, meat pH was unaffected by supplementation in this current goat trial. A study on hemp seed supplementation in Simmental cows also found no significant pH changes between the supplemented and control groups [44]. Similarly, quails fed 0–20% hempseed diets showed no significant pH alterations [45]. However, this study did find lower redness values 1 h after slaughter in goats supplemented with hemp compared with the controlled group. After 24 h, all color values were similar between the two groups, concurring with the cow study [44]. However, redness values of quail breast meat were higher in groups fed hempseed at 5–20% [45]. A review on the absorption and efficacy of hemp by-products in ruminant meat production and preservation [8] found that hempseed oil supplementation can reduce metmyoglobin formation by 10–16% [46] and 18% in vitro [47]. As previously mentioned, fresh hemp leaves reduced the odor of goat meat in this current study. The decreased odor in the supplemented group may be attributed to bioactive compounds in the hemp leaves, particularly phenolics like trans-caffeoyltyramine and cannabisin B, which are abundant in hempseed hulls. These have been documented as having an antioxidant and anti-lipid oxidation effect [48,49,50]. Moreover, maintaining a balanced, thriving gastrointestinal microbiome and fermentation environment may be critical for optimizing quality characteristics like VOCs [51,52]. In addition, a study on the reduction of the formation of undesirable volatile compounds in chickens under inflammation and infection stress showed that dietary CBD could beneficially affect gut health and improve breast meat quality [53].

Concerning the water holding capacity, hemp supplementation numerically decreased drip loss and cooking loss percentages compared to the control group, but this was not statistically significant. The results of this study on water holding capacity, shear values, and texture profile analysis were consistent with a report on hempseed supplementation in Simmental cows, which found no significant differences in cooking loss between the supplemented and control groups [44]. With reference to quails, 20% dietary hempseed supplementation resulted in the lowest cooking loss; however, at other supplementary levels there were no comparable differences [45]. Furthermore, hemp supplementation did not have any significant effect on shear force, shear work, or texture profiles. These findings concur with the cow study in which hemp seed supplementation did not affect the maximum cutting force [44]. The fact that there were no significant differences in the characteristics of meat quality may be attributed to the short study duration, which did not afford enough time for the fresh hemp leaves to influence the textural properties of the goats. Therefore, fresh hemp leaf supplementation did not have a significant impact on the water holding capacity, tenderness, or textural properties of the transported goat meat. However, within the hemp group, some numerical decreases were not statistically significant. More studies are needed to determine whether higher doses of hemp or other preparations could potentially improve the quality of goat meat after transportation. Overall, this current study showed that fresh hemp leaves had minimal effects within these parameters.

The limitations of this study include the brief supplementation and data collection period, the small sample size of only 20 goats, the lack of some physiological measurements during transit to quantify stress levels, the inability to identify specific meat volatiles contributing to the differences in odor, and the lack of sensory analysis to confirm detectable odor changes. Additionally, only one hemp preparation, dose, and duration was investigated. Future research with larger samples over longer timespans should monitor real-time stress biomarkers, utilize advanced techniques to specify influential volatile compounds, incorporate trained sensory panels, and compare multiple hemp by-products, doses, and feeding durations to elucidate optimal benefits for mitigating transportation stress.

## 5. Conclusions

In the present study, a single dose supplementation of 10 g of fresh hemp leaf/kgBW maintained some blood parameters compared to stressed controls. However, growth, carcass yield, and most meat quality properties were not affected by fresh hemp, except for minor differences in initial color and delayed volatile release. Thus, this low dose of fresh hemp leaves provided only limited benefits in alleviating transportation stress in goats. Further research is needed to evaluate higher dietary inclusions of hemp foliage in a greater number of animals to assess its protective potential more comprehensively during livestock transit.

## Figures and Tables

**Table 1 animals-13-03881-t001:** Effects of transportation on blood metabolites of goat treated with fresh hemp leaf.

Items	Control Group				Hemp Group			
	Before	After	SEM	*p*-Value	Before	After	SEM	*p*-Value
White blood cell (×10^3^/mm^3^)	13.55	11.50	0.678	0.148	13.74	10.03	1.517	0.302
Hemoglobin (mg%)	6.58	6.30	0.527	0.061	7.16	6.85	0.587	0.240
Hematocrit (%)	20.25 ^b^	23.25 ^a^	1.713	0.013	21.40	23.75	1.983	0.096
Neutrophil (%)	54.75 ^b^	64.25 ^a^	2.980	0.006	63.60	54.00	2.620	0.063
Lymphocyte (%)	45.00 ^a^	35.75 ^b^	2.964	0.005	34.00 ^b^	45.50 ^a^	2.752	0.035
Eosinophil (%)	0.26	0.00	0.103	NA	2.40	0.50	0.486	0.139
Monocyte (%)	0.00	0.00	NA	NA	0.00	0.00	NA	NA
Blood urea nitrogen	18.96	20.14	0.638	0.404	16.26 ^b^	21.48 ^a^	1.326	0.012
Creatinine	0.91	1.12	0.067	0.161	1.00	0.92	0.034	0.328
Protein total	5.88	5.80	0.115	0.587	5.92	5.92	0.214	1.000
Albumin	2.70	2.76	0.045	0.070	2.88	2.88	0.061	1.000
Globulin	3.18	3.04	0.103	0.311	3.04	3.12	0.176	0.456
Alkaline Phosphatase	62.20	62.60	4.989	0.925	67.00	74.40	9.162	0.355
Sodium	143.54 ^b^	146.82 ^a^	0.770	0.017	145.64 ^b^	148.52 ^a^	1.134	0.045
Potassium	4.73 ^b^	5.96 ^a^	0.298	0.042	5.46	6.08	0.170	0.064
Chloride	109.60 ^b^	112.66 ^a^	0.975	0.007	111.14 ^b^	113.30 ^a^	0.643	0.026
CO_2_	21.40	22.20	0.533	0.242	21.00	20.80	0.567	0.854
Calcium	9.31	9.14	0.170	0.302	9.28	9.05	0.173	0.162
Magnesium	2.23	2.29	0.026	0.236	2.42	2.49	0.054	0.486
Phosphorus	4.76	5.09	0.249	0.617	5.96	6.04	0.348	0.860

Significant changes between the control and hemp-treated groups are shown by letters ^a^ and ^b^ (*p* < 0.05). SEM = standard error of means, NA = not available.

**Table 2 animals-13-03881-t002:** Effects of transportation on performance and some carcass characteristics of goat treated with fresh hemp leaf.

Items	Control	Hemp	SEM	*p*-Value
Initial body weight (kg)	15.78	14.24	0.884	0.414
Final body weight (kg)	15.40	13.97	0.846	0.433
Weight loss (%)	2.39	1.75	0.574	0.607
Carcass weight (kg)	6.36	6.15	0.378	0.805
Carcass percentage (%)	41.33	43.85	0.907	0.177
Fore shank (kg)	0.22	0.19	0.014	0.286
Fore shank (%)	1.44	1.37	0.086	0.724
Hide shank (kg)	0.23	0.23	0.010	1.000
Hide shank (%)	1.53	1.68	0.077	0.336
Head (kg)	1.18	1.11	0.053	0.531
Head (%)	7.72	7.96	0.129	0.374
Penis (kg)	0.19	0.16	0.025	0.547
Penis (%)	1.22	1.12	0.130	0.731
Edible internal organs (kg)	0.73	0.59	0.045	0.141
Edible internal organs (%)	4.75	4.21	0.153	0.076
GI track (kg)	5.36	4.34	0.363	0.174
GI track (%)	34.64	31.04	1.089	0.099
Others (kg)	1.13	1.20	0.062	0.611
Others (%)	7.37	8.76	0.433	0.111

No superscript means no significant group differences (*p* > 0.05).

**Table 3 animals-13-03881-t003:** Effects of transportation on pH, color, and odor characteristics of goats treated with fresh hemp leaf.

Items	1-h Post-Mortem				24-h Post-Mortem			
	Control	Hemp	SEM	*p*-Value	Control	Hemp	SEM	*p*-Value
pH	6.12	6.09	0.068	0.820	6.14	6.32	0.110	0.455
Surface color								
Lightness (L*)	43.22	42.53	1.847	0.864	39.58	40.47	1.362	0.763
Redness (a*)	15.80 ^a^	11.69 ^b^	0.946	0.018	13.53	12.19	0.756	0.405
Yellowness (b*)	6.62	4.66	0.637	0.130	4.74	4.87	0.550	0.911
Sensing response								
Sensor 1	4.00	3.05	0.491	0.359	1.99 ^a^	0.93 ^b^	0.238	0.015
Sensor 2	1.04	1.71	0.257	0.212	3.45 ^a^	2.75 ^b^	0.135	0.001
Sensor 3	8.05	9.78	0.594	0.155	11.41 ^a^	8.90 ^b^	0.474	<0.001
Sensor 4	1.04	0.69	0.202	0.411	0.40	0.36	0.016	0.159
Sensor 5	10.66	11.67	0.548	0.386	10.88 ^a^	8.62 ^b^	0.420	<0.001
Sensor 6	1.01	1.95	0.349	0.194	4.89 ^a^	4.02 ^b^	0.165	<0.001
Sensor 7	0.68	1.44	0.319	0.259	4.47 ^a^	2.78 ^b^	0.391	0.019
Sensor 8	4.09 ^b^	6.26 ^a^	0.490	0.015	1.03	2.04	0.299	0.093

The letters ^a^ and ^b^ (*p* < 0.05) indicate significant differences between groups.

**Table 4 animals-13-03881-t004:** Effects of transportation on water holding capacity, shear values, and texture profile analysis at 24 h p.m. in goat treated with fresh hemp leaf.

Items	Control	Hemp	SEM	*p*-Value
Water holding capacity				
Drip loss (%)	9.13	7.66	0.445	0.100
Cooking loss (%)	18.70	14.64	2.037	0.349
Shear values				
Shear force (kg/cm^2^)	4.86	4.08	0.258	0.135
Work of shear (kg.s)	12.27	12.04	0.456	0.820
Texture profile analysis (TPA)				
Hardness (g)	202.33	218.73	15.110	0.617
Adhesiveness (g.s)	−7.44	−5.44	1.483	0.532
Springiness	0.08	0.09	0.005	0.372
Cohesiveness	0.64	0.67	0.030	0.644
Gumminess	124.48	146.44	13.504	0.449
Chewiness	11.58	16.04	1.970	0.283
Resilience	0.26	0.28	0.021	0.593

No superscript means no significant group differences (*p* > 0.05).

## Data Availability

Data are contained within the article.

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
