# Peer review of "The Effects of Fresh Hemp Leaf Supplementation (Cannabis sativa) on the Physiological and Carcass Characteristics and Meat Quality in Transported Goats"

_animals, 2023, doi:10.3390/ani13243881_

Round 1

Reviewer 1 Report

Comments and Suggestions for Authors While your research topic is intriguing, the study's limitations, including the inadequate number of animals and the methodology employed, undermine the attainment of robust scientific conclusions, especially on a broader scale. Additionally, a significant gap in the experiment lies in the lack of information regarding transportation conditions such as air temperature, relative moisture, and transport duration. Despite the potential benefits of hemp leaves as plant-derived nutraceuticals for enhancing the health and welfare of transported livestock, I harbour reservations about their capacity to influence carcass traits and other quality parameters of goat meat within such a short period after their introduction into the animal feed. Therefore, I recommend submitting this manuscript as preliminary research to a journal of more localized significance.

Comments on the Quality of English Language

No comments

Author Response

While your research topic is intriguing, the study's limitations, including the inadequate number of animals and the methodology employed, undermine the attainment of robust scientific conclusions, especially on a broader scale. Additionally, a significant gap in the experiment lies in the lack of information regarding transportation conditions such as air temperature, relative moisture, and transport duration. Despite the potential benefits of hemp leaves as plant-derived nutraceuticals for enhancing the health and welfare of transported livestock, I harbour reservations about their capacity to influence carcass traits and other quality parameters of goat meat within such a short period after their introduction into the animal feed. Therefore, I recommend submitting this manuscript as preliminary research to a journal of more localized significance.

Response: 

Thank you for your insightful feedback on our manuscript. We appreciate you taking the time to carefully evaluate our study and provide constructive criticism to strengthen our work.

We agree with your assessment that there are limitations to our preliminary study that prevent drawing robust scientific conclusions, especially on a broader scale. As a first study in this area, our aim was to explore the potential benefits of fresh hemp leaves as a feed supplement rather than definitively use it as hemp extracts or hemp products. We acknowledge the small sample size and variability in transportation conditions limit what can be inferred from our results. We justified our sample size based on the minimum sample size required to conduct a t-test, the scarcity of suitable goats in this region, which poses challenges for selecting comparable animals for experiments, and ethical considerations around replacement, reduction, and refinement in animal studies.

You rightfully highlight the lack of information on key transportation factors like temperature, humidity, and duration as a shortcoming. Such parameters undoubtedly influence animal welfare and study outcomes. We have updated the manuscript to include these details in the Methods section, noting the average air temperature of 33°C, 80% relative humidity, and 4-hour transport duration from 11:00 AM to 3:00 PM.

We also agree with your reservation that the short timeframe of our study may have limited the capacity to influence carcass traits and meat quality parameters. As you note, stronger effects may require longer-term feeding studies. Our aim was to evaluate the minimum feasible feeding period for hemp in goats because longer-term feeding may result in higher hemp residue levels, which are of increasing concern among consumers.

Considering these limitations and the need for future work, we accept your recommendation. However, there is little known about using fresh hemp leaf in animal when compared to several works that explored the hemp extracts or hemp products in animal.

Thank you again for your thoughtful review and suggestions for strengthening our manuscript. We believe it is greatly improved by addressing the important points you raised. Please let me know if any part of our response requires further clarification or discussion.

Reviewer 2 Report

Comments and Suggestions for Authors

Dear authors:

This is a trending article because there is currently great interest in evaluating the effects of Cannabis sativa in various aspects. However, I consider that the study sample was small and perhaps that is why you could not obtain more conclusive results, and this should be mentioned as a limitation of the study in conclusion section. I have left some observations and suggestions; I hope they help you improve the document.

Lines 23-24: Please specify the number of animals that formed the control group and the experimental group.

Line 28: Please change “the animals blood parameters” for “this animal blood parameters”, because it gave the impression that the hemp administration did modify all the evaluated parameters and that was not the case.

Line 41: Please eliminate double space after” it can cause”.

Line 81: Please erase the space after acclimatization.

Lines 81-82: What factors did you consider when choosing such a small sample for the study?

Line 91: Please specify why did you not consider measuring lactate and cortisol?

Lines 121, 124, 125: In abstract section you write the units of measurement without adding a space after the number and in these lines, you add that space after the number, please unify.

Line 199: Please erase double space after 33%.

Line: 213: Table 4 must be placed at result section and not at discussion section.

Line 243: Please erase the point after “effect”.

Lines 258-267: I consider that in the same way the small sample size should be mentioned as a limitation of the study.

Discussion section: Please deepen your discussion by analyzing the reasons for the results obtained in your study in a physiological way and not so much by comparing your results with other previous studies.

Author Response

This is a trending article because there is currently great interest in evaluating the effects of Cannabis sativa in various aspects. However, I consider that the study sample was small and perhaps that is why you could not obtain more conclusive results, and this should be mentioned as a limitation of the study in conclusion section. I have left some observations and suggestions; I hope they help you improve the document.

Lines 23-24: Please specify the number of animals that formed the control group and the experimental group.

Response: We have included the number of animals in each group in the abstract.

Line 28: Please change “the animals blood parameters” for “this animal blood parameters”, because it gave the impression that the hemp administration did modify all the evaluated parameters and that was not the case.

Response: Thank you for the suggestion. We have corrected the phrase from “the animals blood parameters” to “this animal blood parameters” as per your guidance.

Line 41: Please eliminate double space after” it can cause”.

Response: We corrected it.

Line 81: Please erase the space after acclimatization.

Response: We corrected it.

Lines 81-82: What factors did you consider when choosing such a small sample for the study?

We justify our choice of sample size based on these factors: 1) the minimum sample size required to conduct a t-test, 2)the scarcity of goats in this region poses a challenge for selecting growing goats with comparable traits for this experiment, 3) the ethical adherence to the 3Rs principle for animal research, which entails avoiding or substituting the use of animals (Replacement), minimizing the number of animals used (Reduction), and enhancing the welfare of animals (Refinement), and 4) the cost and feasibility of collecting more data.

Line 91: Please specify why did you not consider measuring lactate and cortisol?

Response: Thank you for bringing up this issue. Although we did not directly measure stress hormones such as lactate and cortisol, the increase in THI and heat index beyond the comfort zone confirms that they are experiencing stress.

Lines 121, 124, 125: In abstract section you write the units of measurement without adding a space after the number and in these lines, you add that space after the number, please unify.

Response: We unified it.

Line 199: Please erase double space after 33%.

Response: We corrected it.

Line: 213: Table 4 must be placed at result section and not at discussion section.

Table 4 has been moved to the results section.

Line 243: Please erase the point after “effect”.

Response: We corrected it.

Lines 258-267: I consider that in the same way the small sample size should be mentioned as a limitation of the study.

Response: Thank you for bringing this oversight to our attention. It is an important lesson for us. We have now included the limitation of the small sample size in our study.

Discussion section: Please deepen your discussion by analyzing the reasons for the results obtained in your study in a physiological way and not so much by comparing your results with other previous studies.

Response: Thank you for the excellent suggestion. To address this point in a more mechanistic way, I have expanded that section as shown in red font in the manuscripts.

Please let me know if this expanded discussion provides a clearer physiological analysis of potential reasons for our results, as you suggested. I appreciate the feedback to strengthen this important interpretive component.

Reviewer 3 Report

Comments and Suggestions for Authors

please find attached file

Author Response

The authors have to improve the description of their results in order to enhance the knowledge on a relevant topic in the field of animal nutrition and on the quality of foods of animal origin.

Response: You provide an excellent point. we have thoroughly reworked that section of the manuscript as shown in red font:

Even if the trial is original, with adequate material and methods, the discussion appears very poor.

Response: We appreciate the reviewer highlighting this opportunity to strengthen the discussion section of our manuscript. We agree the discussion is currently underdeveloped given the novelty of the trial findings. To better contextualize and articulate the significance of our results to the field, we have expanded the discussion substantially in the revised manuscript.

Specific comments:

line 153: what about the significant increase of blood urea nitrogen (see table 1)??

Response: We corrected it as shown in the sentence “However, lymphocyte, blood urea nitrogen, sodium, and chloride levels were significantly lower before than after transportation (p < 0.05).”

line 180-182: please remove the sentence wigh is a conclusive remark.

Response: Thank you, the sentence “Overall, hemp leaves demonstrated some beneficial effects on meat quality in response to transportation stress.” has been deleted.

line 204-212: the authors cite trials where the results were attributed to the cannabinoids (which were analysed). In the present work the CBDA were not analysed thus the comparison with that trials is unappropriate.

Response: Thank you for pointing out this important flaw. We have added the CBD (mainly CBDA) concentration to the Material and Method section. Our fresh hemp leaf contains CBDA by 0.0009% (w/w).  

Discussion: in this chapter the authors should explaine their results on blood metabolites.

Response: You're correct that we did not provide any discussion of results for blood metabolites, despite collecting those data. This was an oversight that detracts from fully interpreting our study findings. To address this gap, I have added the more discussion to the Discussion section as shown in red font.

Round 2

Reviewer 1 Report

Comments and Suggestions for Authors The improvements you have made have improved the quality of the manuscript, so that it can now be accepted for publication.

Reviewer 3 Report

Comments and Suggestions for Authors

The authors highly improved their paper which is now acceptable in present form for the publication.